# Case–Control Study: Endogenous Procalcitonin and Protein Carbonylated Content as a Potential Biomarker of Subclinical Mastitis in Dairy Cows

**DOI:** 10.3390/vetsci10120670

**Published:** 2023-11-24

**Authors:** Giulia Sala, Chiara Orsetti, Valentina Meucci, Lucia De Marchi, Micaela Sgorbini, Francesca Bonelli

**Affiliations:** 1Department of Veterinary Sciences, University of Pisa, Via Livornese s.n.c., San Piero a Grado, 56122 Pisa, Italy; chiara.orsetti@phd.unipi.it (C.O.); valentina.meucci@unipi.it (V.M.); lucia.demarchi@unipi.it (L.D.M.); micaela.sgorbini@unipi.it (M.S.); francesca.bonelli@unipi.it (F.B.); 2Centro di Ricerche Agro-Ambientali “E. Avanzi”, University of Pisa, San Piero a Grado (PI), 56122 Pisa, Italy

**Keywords:** subclinical mastitis, biomarker, procalcitonin, protein carbonylated content

## Abstract

**Simple Summary:**

This case–control study investigated procalcitonin (PCT) and protein carbonylated content (PCC) during subclinical mastitis in dairy cows. A total of 130 cows (65 healthy and 65 with subclinical mastitis) were examined. This study revealed a significant difference in PCT levels between healthy and subclinical mastitis cows. However, there was an unexpected trend in PCC concentrations. This study established a PCT cutoff value of >89.8 pg/mL for distinguishing between healthy and subclinical mastitis animals, with a sensitivity of 66.2% and specificity of 69.2%. PCT showed potential value as a diagnostic tool to help in decision making for subclinical mastitis cases, while PCC requires further studies to investigate the trend of this biomarker during localized pathology.

**Abstract:**

Procalcitonin (PCT) and protein carbonylated content (PCC) are promising biomarkers for bacterial infection and inflammation in veterinary medicine. This study examined plasma PCT and PCC levels in healthy cows (H) and cows with subclinical mastitis (SCM). A total of 130 cows (65 H and 65 SCM) were included in this study. Blood samples were collected, and plasma was frozen at −80 °C. PCT levels were determined using a bovine procalcitonin ELISA kit, while PCC was measured following the methodology of Levine et al. Statistical analysis revealed a significant difference in PCT levels between H (75.4 pg/mL) and SCM (107.3 pg/mL) cows (*p* < 0.001) and significantly lower concentrations of PCC in the SCM group (H: 0.102 nmol/mL/mg, SCM: 0.046 nmol/mL/mg; *p* < 0.001). The PCT cut-off value for distinguishing healthy and subclinical mastitis animals was >89.8 pg/mL (AUC 0.695), with a sensitivity of 66.2% and specificity of 69.2%. PCT showed potential value as a diagnostic tool to help in decision making for subclinical mastitis cases, while PCC requires further studies to investigate the trend of this biomarker during localized pathology.

## 1. Introduction

Biomarkers in veterinary medicine are used to assess the health status of animals, to make a diagnosis, to predict disease progression, and to evaluate the effectiveness of treatments [1,2].

In veterinary medicine, procalcitonin (PCT) has emerged as a significant biomarker among various biomarkers showing promising results across different species and diseases [3,4,5,6]. PCT, a glycoprotein consisting of 116 amino acids, is the precursor of the hormone calcitonin [7]. Under normal physiological conditions, PCT is primarily synthesized by thyroid C cells and immediately converted to calcitonin, resulting in minimal release into the bloodstream [8,9]. However, in human medicine, it was demonstrated that the production of PCT can significantly increase, ranging from 100 to 1000 times, during systemic inflammatory response syndrome (SIRS) and sepsis [9]. The elevated levels of PCT during diseases can be attributed to the overexpression of the CALC-1 gene, responsible for protein synthesis. Research in human medicine has shown that this extra-thyroid synthesis of PCT occurs in various tissues, including the liver, pancreas, kidneys, lungs, intestines, and leukocytes [10]. This extra-thyroid synthesis of PCT is promoted in the presence of bacterial infection, while during viral infections, released cytokines such as interferon-gamma induce a down-regulation of PCT [11]. For this reasons, PCT is used in human medicine as a marker to guide antibiotic treatment in different diseases [12,13,14,15]. In veterinary medicine, PCT was also found to be a reliable biomarker for various diseases condition. Several studies have been conducted in large animals using PCT as a biomarker of bacterial infections, for example, in adult horses [4,16,17,18,19], foals [20,21], adult cows [22,23,24], and calves [25,26,27,28,29,30]. In all these studies, plasma PCT levels have demonstrated the ability to distinguish between sick and healthy animals, but the deep knowledge that PCT brings for human medicine is still missing in veterinary species.

Another emerging biomarker of oxidative stress during diseases is the content of carbonylated proteins (PCC). Inflammatory conditions can trigger oxidative stress when there is an imbalance in the generation of reactive oxygen species (ROS) and the antioxidant activity within the body, resulting in the accumulation of ROS that directly damage cells and tissues [31]. ROS (such as superoxide, peroxyl, hydroxyl, and hydrogen peroxide), are highly energetic and reactive small molecules derived from oxygen [32]. Moreover, oxidative stress triggers the activation of pro-inflammatory cytokines, leading to subsequent inflammation that further amplifies ROS production, causing additional damage to cells and tissues [33]. ROS are involved in the introduction of carbonyl groups (aldehyde and ketone) into protein side chains through various oxidation reactions [34]. Normally, proteins in healthy tissues have low levels of carbonyl groups [35]. Measuring the content of protein carbonyls allows for the assessment of oxidative modifications [36] and quantification of oxidative stress, also associated with inflammation [37]. Carbonylated proteins have an extended half-life, making the evaluation of carbonyl group content a valuable indicator of the extent of oxidative stress in disease conditions [37]. In veterinary medicine, there are a few studies about PCC that have shown an increased concentration of plasma PCC in horses with systemic inflammatory response syndrome [38], buffalos with *Theileria annulata* [39], and dogs with sepsis [40,41]. In bovine medicine, PCC concentration was evaluated in the milk of healthy animals and dairy cattle with mastitis [42,43].

Mastitis, an inflammatory condition of the mammary gland, involves bacterial infection and, in cases of antioxidant defense imbalance, inflammatory oxidative stress. Various pathogens, including bacteria, viruses, fungi, and algae, can induce mastitis, leading to tissue damage [44] and economic losses in the dairy industry [45]. Mastitis manifests in two forms: clinical mastitis, characterized by visible changes in the milk and udders, and subclinical mastitis, which lacks macroscopic alterations in the milk but is associated with reduced milk production, altered milk composition, and an elevated somatic cell count (SCC) [46]. Subclinical mastitis is notably more prevalent, with a prevalence 15–40 times higher than clinical mastitis [47]. Detecting subclinical mastitis poses a challenge due to the absence of observable clinical indicators. The most commonly used method for diagnosing subclinical mastitis is measuring the somatic cell count (SCC) in the milk, detectable through ancillary tests [48]. Diagnosis typically relies on assessing milk SCC, using a threshold of 200,000 cells/mL [49]. Other methods, albeit less accurate ones, include milk lactose concentration, milk enzyme levels (lactate dehydrogenase and N-acetyl-b-D-glucosaminidase), acute-phase protein concentration in milk (haptoglobin and amyloid A), and milk electrical conductivity [50].

The aim of this study was to assess potential variations in plasma concentrations of PCT and PCC between healthy animals and those with subclinical mastitis. The hypothesis is that mammary infection and increased oxidative stress at the mammary level may lead to modifications in these biomarkers at the systemic level. Furthermore, this study has an additional objective, which is to determine the diagnostic accuracy of these biomarkers in cases of subclinical mastitis.

## 2. Materials and Methods

### 2.1. Study Design

In adherence to the STROBE guidelines, we performed a case–control investigation at the dairy farm of the University of Pisa (Centro di Ricerche Agro-Ambientali “E. Avanzi”). The large animals service of the Department of Veterinary Sciences (DSV) of the University of Pisa is in charge of the monitoring of the udder health of the farm. This study was approved by the Institutional Animal Care and Use Committee of the University of Pisa (prot. N: 2825 of 28 January 2014) and written consent from the owner was obtained for the inclusion of all cows in this study.

### 2.2. Animals and Management

Holstein Friesian lactating cows underwent the same management practices. The cows were accommodated in a free-stall barn where straw was utilized as bedding material. The bedding was replaced twice a week, with additional clean straw added daily. Each lactating cow was provided with the same total mixed ration twice a day, along with free access to fresh water. Lactating cows were milked twice a day using a Herringbone milking parlor, with an interval of approximately 11 h between milkings (at 5 a.m. and 4 p.m.). On this farm, a veterinary weekly check is performed during the afternoon milking session to monitor the udder health and the teat condition of all lactating cows.

### 2.3. Inclusion Criteria

During udder health monitoring carried out by the large animals service of the DSV, all the Holstein Friesian lactating cows underwent somatic cell count (SCC) evaluation at quarter level and a clinical examination. Cows were assigned to the subclinical mastitis (SCM) group based on the following inclusion criteria: (1) having an SCC > 200,000 cells/mL in at least one milk quarter, without any observable changes in the udders or macroscopic abnormalities in the milk [49]; (2) no occurrence of clinical mastitis or subclinical mastitis during the lactation period under examination; and (3) absence of any other concurrent diseases. The inclusion criteria for the healthy (H) group were as follows: (1) having an SCC < 200,000 cells/mL with no observed alterations in the udders or milk [49]; (2) no occurrence of clinical or subclinical mastitis during the examined lactation period; and (3) absence of any other underlying pathological conditions.

### 2.4. Collection of Samples

A milk quarter sample and a blood sample were taken from each cow included in the study, for both the H and SBM groups. Milk samples were obtained at the time of subclinical mastitis diagnosis, following the National Mastitis Council guidelines [51]. Before sampling, a thorough cleaning of the teat was conducted using a pre-dipping foam containing lactic acid (Biofoam Plus, DeLaval Inc., Tumba, Sweden). The teat was then dried, and the apex was disinfected using alcohol [52]. The initial streams of foremilk were discarded, and approximately 10 mL of milk was aseptically collected from each teat in sterile vials. The collected samples were stored at 4 °C until the bacteriological assays were conducted. For the bacteriological analysis, ten microliters of each milk sample were spread on 5% defibrinated sheep blood agar plates. After 24 h aerobic incubation at 37 °C, colonies were provisionally identified based on Gram staining, morphology, and hemolysis patterns. Selected colonies were subcultured on fresh blood agar plates. Gram-positive cocci were tested for catalase and coagulase production, while Gram-negative isolates were identified using colony morphology, Gram staining, oxidase tests, and MacConkey agar biochemistry. Samples with three or more pathogens were considered contaminated. The SCC was measured immediately after sampling using an automated somatic cell counter (DCC, DeLaval International AB, Tumba, Sweden). The blood samples were collected from the coccygeal veins in lithium heparin tubes and immediately centrifuged at 2100× *g* for 10 min. The resulting plasma was carefully transferred to sterile tubes and stored in a freezer at −80 °C until further analysis. To preserve the bioactivity of the samples, they were defrosted on ice for approximately 2 h before use and analyzed in a single batch within 6 months.

### 2.5. Determination of Plasma PCT and PCC Concentration

The analysis of plasma PCT and PCC were conducted at the Veterinary Pharmacology and Toxicology Laboratory, located in the Department of Veterinary Sciences at the University of Pisa. The concentration of PCT in the plasma samples was determined using a commercial kit for cattle (Bovine Procalcitonin ELISA kit, Cusabio, Houston, TX, USA), as already described in Meucci et al. [53]. To ensure the reliability of the assay, the intra-assay and inter-assay coefficients of variation were established, and variations were found to be less than 20%. The limit of detection of the method, as indicated by the manufacturer, was 40 pg/mL. In order to validate the specified detection limit for bovine plasma PCT, we conducted a dilution series measurement of PCT using bovine samples with low PCT concentrations (<40.0 pg/mL). The results falling below the detection limit were carefully examined for confirmation, subsequently reported, and included in the statistical analysis as lod/2.

PCC was assessed following the methodology of Levine et al. [54]. This method involves measuring the reaction between 2.4-dinitrophenyl hydrazine and protein carbonyls to generate protein hydrazone. Briefly, for each sample, a protein solution (1–10 mg/mL) was pipetted into 1.5 mL centrifuge tubes. To each tube, 500 µL of 10 mM 2,4-dinitrophenylhydrazine (DNPH) in 2 M HCl was added and allowed to stand at room temperature for 1 h. The samples were then centrifuged at 11,000× *g* for 3 min after the addition of 500 µL of 20% trichloroacetic acid (TCA). The obtained pellets were washed three times with 1 mL of ethanol–ethyl acetate (1:1) to remove free reagent, allowing the sample to stand for 10 min before each centrifugation, and discarding the supernatant each time. The precipitated protein was then redissolved in 0.6 mL of guanidine solution. Any insoluble materials were removed through centrifugation for 3 min. The resulting hydrazones were quantified spectrophotometrically at 370 nm absorbance (Synergy™ HTX, Bio Teck Instruments Incorporated, Winooski, VT, United States). The PCC was calculated using the molar absorption coefficient of 22,000 M^−1^ cm^−1^ relative to protein concentration and expressed as nmol/mL/mg of total protein. The total protein content was determined using the Lowry method [55] with bovine serum albumin (BSA) as the standard. The Lowry method is a widely used colorimetric assay for determining the protein concentration in a sample. It is based on the reaction of proteins with copper ions (Cu^2+^) in an alkaline solution, resulting in the formation of a purple-colored complex that can be measured spectrophotometrically at 750 nm. In a nutshell, each diluted sample (1 mL) was pipetted into a test tube, and an equal volume of Folin–Ciocalteu phenol reagent was added. The contents were gently mixed and incubated for 10 min to allow color development. After the incubation, copper sulfate solution was added to each tube. The solution was mixed again and incubated for 30 min to allow the formation of a purple-colored complex. The absorbance was measured at 750 nm, and the content was expressed as mg/mL. The intensity of the purple color is directly proportional to the protein concentration.

### 2.6. Statistical Analysis

The sample size was determined using G-power software (Ver. 3.1, Heinrich-Heine-Universität, Düsseldorf, Germany) and calculated with a Wilcoxon–Mann–Whitney test. For the calculation, an effect size of 0.5 (medium), a type I error (α) of 5%, a confidence interval of 95%, and a test power of 80% were utilized. The minimum number of animals required was 53 per group.

Descriptive statistic and non-parametric tests were performed with IBM SPSS Statistics v. 27.0 (IBM Corp., Armonk, NY, USA). The quantitative variables collected were age, DIM, PCT, and PCC concentration, while the qualitative variables were presence/absence of subclinical mastitis, bacteriological results, cow parity, and body condition score (BCS). Data distribution was checked with the Shapiro–Wilk test, and the result was non-normal distribution. For this reason, the descriptive statistics of quantitative variables were reported as the median and 25% and 75% percentile, while qualitative variables were reported as frequencies and percentage.

The medians of age, DIM, PCT, and PCC in cows with or without subclinical mastitis were compared using the Mann–Whitney U test, while the frequencies of cow parity and BCS were analyzed with the chi-square test. To evaluate the potential impact of DIM and cow parity on PCT and PCC, the biomarker analysis was conducted using the Kruskal–Wallis test and Bonferroni’s post hoc test across different categories of cow parity and DIM (early lactation = 0–100 days; mid-lactation = 101–200 days; late lactation = >201). Statistical significance was considered for a *p*-value < 0.05.

If a difference between the healthy and pathological groups was detected, diagnostic accuracy analysis was performed with MedCalc^®^ Statistical Software version 22.003 (MedCalc Software Ltd., Ostend, Belgium; https://www.medcalc.org; accessed on 23 September 2023). Cut-offs were determined with the receiver operating characteristic (ROC) curve, and the cut-off was chosen using the Younden index (J) [56], where sensitivity and specificity are maximized and equal weight is given to false-positive and false-negative results (J = Sensitivity + Specificity − 1). In addition, the areas under the curve (AUC) and their 95% confidence intervals (CIs) were calculated and used as indicators of test accuracy to discriminate the subclinical mastitis. The cut-off values selected were used to estimate sensitivity (Se), specificity (Sp), positive predictive value (PPV), and negative predictive value (NPV).

## 3. Results

During the study period, a total of 174 cows were in lactation. Nineteen animals were not eligible for the study in either the H group or the SCM group because they showed diseases other than SCM. Among the eligible cows, 75 developed subclinical mastitis. However, five cows were excluded due to experiencing previous clinical mastitis in the same lactation period, two cows were excluded due to lameness, and three cows were excluded due to metritis. This resulted in an SCM group consisting of 65 animals. Additionally, there were 80 cows with optimal udder health during the study. However, seven cows were excluded due to previous subclinical mastitis in the same lactation period, three cows were excluded due to previous clinical mastitis in the same lactation period, two cows were excluded due to metritis, two cows were excluded due to lameness, and one cow was excluded due to lumpy jaw disease. This resulted in an H group consisting of 65 animals.

The descriptive statistics for the SCM and H groups are reported in Table 1 and Table 2, respectively, along with the results of non-parametric testing. The bacteriological analysis results were positive in 18 (27.7%) cases of subclinical mastitis, and negative in the remaining cases of subclinical mastitis (47/65, 72.3%) and in all healthy animals (65/65, 100%). The most present bacteria were Staphylococcus spp. (8/18, 44.4%), followed by Aerococcus viridans (4/18, 22.2%), Enterobacteriacie (3/18, 16.7%), Enterococcus spp. (2/18, 11.1%), and Streptoccocus spp. (1/18, 5.6%).

The median age in the H group was 3 years old (2 years old–6 years old), while in SCM group, it was 4 years old (3 years old–6 years old). For PCT, the median concentration in the H group was 75.4 pg/mL (37.4 pg/mL–99.1 pg/mL), whereas in the SCM group, it was 107.3 pg/mL (68.6 pg/mL–253.3 pg/mL), showing a statistically significant difference (*p*-value < 0.001, Figure 1). Regarding PCC, the median concentration in the H group and SCM group was 0.102 nmol/mL/mg (0.038 nmol/mL/mg–0.193 nmol/mL/mg) and 0.046 nmol/mL/mg (0.021 nmol/mL/mg–0.095 nmol/mL/mg), respectively, with a statistically significant difference (*p*-value < 0.001, Figure 1). The biomarker analysis conducted across different categories of cow parity (PCT *p*-values = 0.238; PCC *p*-value = 0.378) and DIM (PCT *p*-values = 0.768; PCC *p*-value = 0.593) did not reveal any statistically significant differences.

The ROC curve for PCT is reported in Figure 2. The cut-off resulting from ROC analysis was >89.8 pg/mL, with an AUC of 0.695 (CI 0.61–0.79) and a J index of 0.35. The Se and Sp were 66.2% (CI 53.4–77.4%) and 69.2% (CI 56.6–80.1%), respectively. The PPV and NPV were 68.25% (CI 58.9–76.3%) and 67.2% (CI 58.4–74.9%), respectively.

The ROC curve for PCC is reported in Figure 3. The cut-off resulting from ROC analysis was ≤0.066 nmol/mL/mg, with an AUC of 0.678 (CI 0.59–0.76) and a J index of 0.32. The Se and Sp were 68.8% (CI 55.9–79.8%) and 63.1% (CI 50.2–74.7%), respectively. The PPV and NPV were 64.7% (CI 56.2–72.4%) and 67.2% (CI 57.7–75.5%), respectively.

## 4. Discussion

The results of our study revealed an interesting finding, indicating that PCT can effectively differentiate healthy cows from those with subclinical mastitis. These results are in line with the existing literature on calves [25,26,27], cows [22,23,24], horses [4,16,17,18,57], and foals [20]. In humans, it has been observed that PCT increases in response to stimulation by TNF-α and IL-1β [7]. In cattle, the direct association between PCT and these cytokines has not yet been demonstrated, but the specific elevation of these two cytokines during subclinical mastitis has been highlighted by Shaheen et al. [58]. This would explain how PCT can differentiate healthy animals from those affected also by a localized pathology. The impact of localized conditions such as subclinical mastitis on other proteins has also been highlighted by other proteomic studies, both in serum and milk [59]. This further supports the notion that subclinical mastitis has systemic effects.

The optimal cut-off value identified in our study (89.8 pg/mL) was slightly higher than that reported for neonatal calves [26], feedlot calves [27], and cows with clinical staphylococcal mastitis [22] (67.39, 48.62, and 56.16 pg/mL, respectively). However, it was lower than values reported for cows hospitalized for bacterial infection (244.44 pg/mL) [23] and clinical and subclinical mastitis (2641 pg/mL and 1961 pg/mL, respectively) [24]. The observed variations in PCT levels between our study and the previous literature can be attributed to several factors, including differences in PCT determination methodologies [23,24], variations in subject age [26,27], and variations in the underlying diseases [23,26,27]. The variation resulting from the different analysis methods is particularly evident between our study and the study by Neumann et al. [24]. Indeed, in this study, a different ELISA kit was used, and it is the only study in cattle that used serum as the matrix. A potential explanation for the differences between our study and that of Bonelli et al. [23] lies in the specific diseases investigated. Our study focused on subclinical and localized disease, which triggers a limited systemic response [60], while in Bonelli et al.’s [23] study, the cows included were affected by diseases that required hospitalization. In contrast, El-Deeb et al. [22] examined mastitis associated only with Staphylococcus aureus infection, and it is well documented in human medicine that PCT levels vary depending on the etiological agent [13]. Specifically, Gram-negative bacteria induce a more pronounced increase in PCT levels than Gram-positive bacteria [13,61]. In our study, the bacterial examination identified mixed microbial flora, potentially explaining the differences in PCT values between the diseased animals in our study and those in the study by El-Deeb et al. [22].

The mean sensitivity and specificity achieved at the best cut-off value was acceptable (66.2% and 69.2%, respectively). These results were comparable to those found in cows hospitalized for bacterial infection (Se 73.6% and Sp 60%) [23] and better than those observed in cows with subclinical mastitis in Neumann et al.’s study (Se 66%; Sp 35%) [24]. Previous studies in horses, calves, and cows have reported sensitivity and specificity for the determined cut-off values ranging from 81% to 100% and from 69% to 97.1%, respectively [18,19,20,26,27]. Interestingly, in a study that examined PCT in cases of mastitis associated with Staphylococcus aureus, the diagnostic accuracy of PCT was considerably higher when bacterial involvement was confirmed. PCT demonstrated a sensitivity and specificity of 100% in distinguishing between healthy subjects and those with mastitis [22]. This suggests that the accuracy of PCT in distinguishing between healthy cows and those with mastitis increases when there is an identifiable bacterial infection present. The observed decrease in diagnostic accuracy in our study can potentially be attributed to the low number of positive bacteriological examinations among the samples. Out of the 65 samples analyzed, only 18 tested positive for bacterial infection. It is important to note that PCT is primarily recognized as a marker for bacterial infections and is extensively used in human medicine to guide antibiotic treatment decisions, including the duration of treatment [14,15].

Another factor that may have influenced the difference in diagnostic accuracy between our study and the study of El-Deeb et al. [22] is that we focused exclusively on subclinical mastitis cases, excluding those with clinical mastitis, and bacterial isolation was not performed for all cases of subclinical mastitis. Clinical mastitis typically exhibits more pronounced clinical manifestations and bacterial involvement [46,62]

Further research is needed to investigate the diagnostic potential of PCT in mastitis cases with different causative agents and varying degrees of severity. This knowledge will contribute to informed decision making regarding the use of PCT as a biomarker for guiding antibiotic treatment, similar to its application in human medicine.

The result regarding PCC was unexpected. PCC effectively differentiates healthy animals from cows with subclinical mastitis, but the concentration of PCC decreases in sick animals. This finding is not in line with the literature. One study [63] in dairy cows found no significant differences in the concentration of carbonylated proteins in healthy subjects and subjects with subclinical and clinical endometritis, and other studies conducted on dogs with systemic inflammatory response syndrome (SIRS) [40,41,64], horses with SIRS [38], calves parasitized by *Theileria annulata* [39], and cows with mastitis [42,43] demonstrated statistically increased PCC concentrations between the pathological subjects and healthy control groups.

In the literature, some studies show that the blood protein profile of cows with subclinical mastitis differs from that of healthy animals [59,65,66,67,68]. Studies examining plasma PCC concentration typically express results relative to the total protein concentration of plasma [69]. Measuring protein carbonylation in this way does not account for possible changes induced by the disease in the protein profile of cows with subclinical mastitis. This is relevant because the susceptibility of proteins to oxidation varies [70]. Therefore, differences in the protein profile between healthy animals and cows with subclinical mastitis can influence plasma PCC levels. Furthermore, in cases of subclinical mastitis, where the alterations are primarily localized, oxidative stress may not significantly impact the systemic level. Therefore, a possible explanation for the reduction in PCC concentration is that the change in the blood protein profile in cows with subclinical mastitis and the simultaneous localization of the inflammatory process in the udders reduced the plasma PCC concentration in our study. The differences in DIM between the H and SCM groups in our study did not influence the concentration of PCC. This result was also unexpected. In fact, Kuhn et al. [71] demonstrated that oxidative status changes during lactation, with an increase in oxidation in early lactation that decreases during mid- and late lactation. However, PCC was not investigated as a marker of oxidative status. Further studies are therefore necessary to investigate PCC in both healthy animals during various stages of lactation and in the case of subclinical mastitis.

Hence, the results regarding the diagnostic accuracy of PCC in our study should be interpreted with caution, and further studies are needed to confirm or refute the explanation for the reduction in PCC.

Additionally, our study has limitations. Blood tests were not conducted for the purpose of this study. However, the likelihood of including unhealthy animals in the control group is considered very low, as the herd from which the animals come undergoes weekly screening for various diseases and/or clinical/subclinical conditions by the veterinary service of the institution. Another limitation is that PCT and PCC were measured only in plasma and not in milk. Furthermore, the number of animals included should be increased, especially considering the potential role of etiology of mastitis in biomarkers.

## 5. Conclusions

PCT showed potential value as a diagnostic tool to help in decision making for subclinical mastitis cases. However, for widespread adoption of this biomarker, the development of a cheap and commercially available kit is essential. Future studies should include both PCT and PCC in plasma and milk, considering various degrees of mastitis severity and different etiological causes, to determine appropriate cut-off values.

## Figures and Tables

**Figure 1 vetsci-10-00670-f001:**
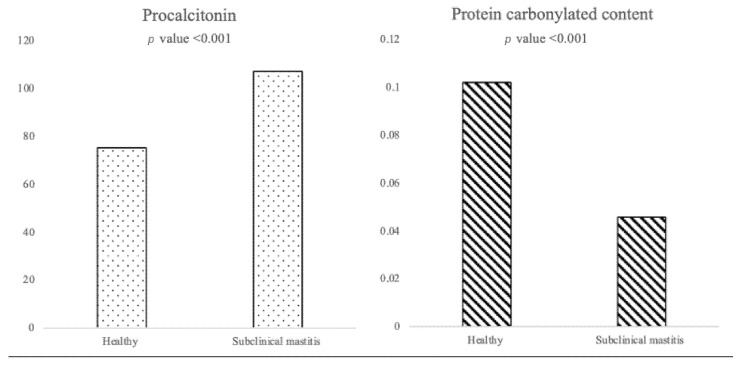
Results of the Mann–Whitney U test of procalcitonin (PCT) and protein carbonylated content in healthy (H) and subclinical (SCM) groups. The median PCT concentration in the H group was 75.4 pg/mL, whereas in the SCM group, it was 107.3 pg/mL. The PCC median concentration in the H group and SCM group was 0.102 nmol/mL/mg and 0.046 nmol/mL/mg, respectively.

**Figure 2 vetsci-10-00670-f002:**
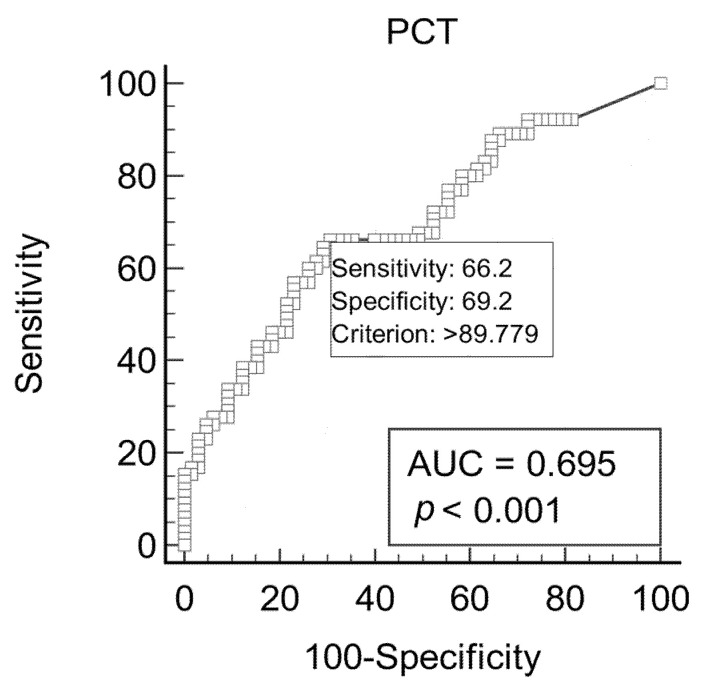
Receiver operating characteristic (ROC) curves and area under the curve (AUC) for optimal procalcitonin (PCT) cut-off in 130 cows to discriminate the presence or absence of subclinical mastitis.

**Figure 3 vetsci-10-00670-f003:**
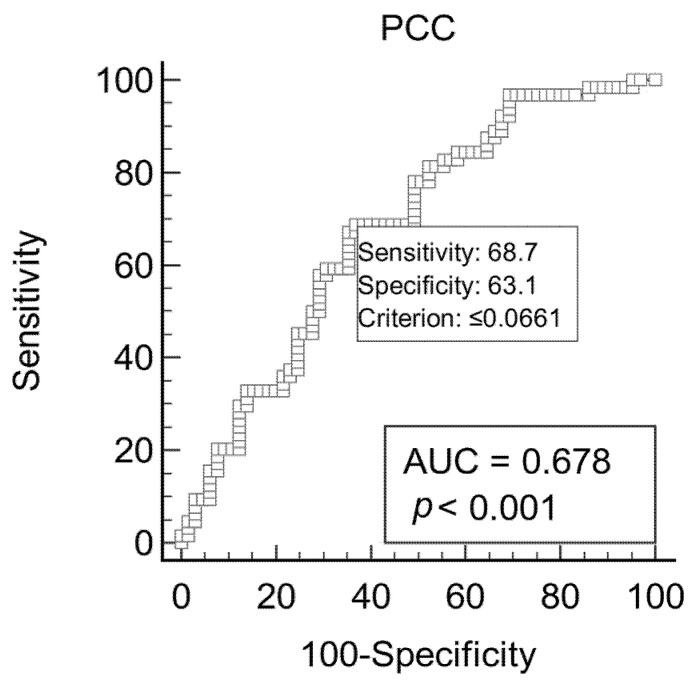
Figure 2: receiver operating characteristic (ROC) curves and area under the curve (AUC) for optimal protein carbonylated content (PCC) cut-off in 130 cows to discriminate the presence or absence of subclinical mastitis.

**Table 1 vetsci-10-00670-t001:** Descriptive statistics of plasma procalcitonin (PCT), protein carbonylated content (PCC), and daily in milk (DIM) split for healthy cows and cows with subclinical mastitis. These data are reported as median (25% percentile–75% percentile).

	PCT (pg/mL) *	PCC (nmol/mL/mg) *	DIM (Days) *
Healthy	75.36 (37.44–99.14)	0.102 (0.038–0.192)	60.00 (15.00–150.00)
Subclinical mastitis	107.27 (68.63–253.25)	0.046 (0.021–0.095)	170.00 (58.75–290.75)

* indicates a *p*-value < 0.05 determined with the Mann–Whitney U test between the H group and the SCM group.

**Table 2 vetsci-10-00670-t002:** Descriptive statistics of body condition score (BCS) and cow parity split for healthy cows and cows with subclinical mastitis. These data are reported as frequencies (percentage).

	Healthy	Subclinical Mastitis
BCS		
2	1 (1.5%)	/
2.5	1 (1.5%)	/
2.75	7 (10.8%)	9 (13.8%)
3	35 (53.9%)	30 (46.2%)
3.25	12 (18.5%)	12 (18.5%)
3.5	6 (9.2%)	8 (12.3%)
3.75	3 (4.6%)	5 (7.7%)
4	/	1 (1.5%)
Cow parity *		
1	30 (46.2%)	21 (32.3%)
2	30 (46.2%)	14 (21.5%)
>3	5 (7.6%)	30 (46.2%)

* indicates a *p*-value < 0.05 determined with the chi-square test between the H group and the SCM group.

## Data Availability

The datasets generated and analyzed during the current study are available from the corresponding author upon reasonable request.

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
