# Peer review of "Case–Control Study: Endogenous Procalcitonin and Protein Carbonylated Content as a Potential Biomarker of Subclinical Mastitis in Dairy Cows"

_vetsci, 2023, doi:10.3390/vetsci10120670_

Round 1
Reviewer 1 Report
Comments and Suggestions for Authors
The authors described plasma procalcitonin and protein carbonylated content in healthy cows and those with subclinical mastitis. This topic is relevant to the field of inflammatory biomarkers in high producing dairy cows.
There are some points that should be addressed:
Introduction and Discussion:
The mechanism of procalcitonin action should be better explain. More about this should be discussed in the Discussion section as well, particularly in the light of results of other authors who investigated different inflammatory proteins in subclinical and clinical mastitis (discuss the paper Milk and serum proteomes in subclinical and clinical mastitis in Simmental cows doi.org/10.1016/j.jprot.2021.104277 and similar). The relevance of oxidative stress in relation to inflammation should be more described and the process of PCC formation as well. The unexpected PCC results should be more discussed in the Discussion section and the relevance why this parameter is included in the study.
Material and methods:
Cows’ breed and age should be indicated.
The method for PCC measurement needs to be more explained in details. The manufacturer for spectrophotometer for PCC determination should be indicated as well as how the quality control for this assay was performed.
Results:
Please indicate statistical significant differences in the Tables 1 and 2 by asterisk.
Author Response
The authors described plasma procalcitonin and protein carbonylated content in healthy cows and those with subclinical mastitis. This topic is relevant to the field of inflammatory biomarkers in high producing dairy cows.
Au: We want to express our thanks to the reviewer, whose comments have contributed to improving our paper.
There are some points that should be addressed:
Introduction and Discussion:
The mechanism of procalcitonin action should be better explain. More about this should be discussed in the Discussion section as well, particularly in the light of results of other authors who investigated different inflammatory proteins in subclinical and clinical mastitis (discuss the paper Milk and serum proteomes in subclinical and clinical mastitis in Simmental cows doi.org/10.1016/j.jprot.2021.104277 and similar). The relevance of oxidative stress in relation to inflammation should be more described and the process of PCC formation as well. The unexpected PCC results should be more discussed in the Discussion section and the relevance why this parameter is included in the study.
AU: Thanks for the suggestion. The information required was added in the introduction (see lines 42-67 and line 74-81) and in the discussion (see lines 472-479 and lines 732-738).
Material and methods:
Cows’ breed and age should be indicated.
AU: thanks for the comment. The animals included in the study were Holstein Friesian cows (see lines 212 and 222). Age was not an inclusion criterion, and the age of the animals has been added to the results section. See lines 417-418
The method for PCC measurement needs to be more explained in details. The manufacturer for spectrophotometer for PCC determination should be indicated as well as how the quality control for this assay was performed.
AU: Thanks for the suggestion. The method used for the PCC analysis was rewrite with more detail at lines 264-301.
Results:
Please indicate statistically significant differences in the Tables 1 and 2 by asterisk.
AU: thanks for the comment. The information was added in the tables.
Reviewer 2 Report
Comments and Suggestions for Authors
Thank you for the opportunity to review the manuscript by Sala et al. Case-Control study: endogenous procalcitonin and protein carbonylated content as a biomarker of subclinical mastitis in dairy cow. This manuscript compares the plasma values of PCT and PCC in both healthy cows and those with sub-clinical mastitis across parities and periods of lactation. The findings were positive for PCT and negative for PCC, compared to the proposed hypothesis, but it is VERY appreciated that the negative data was presented thoroughly and discussed.
Overall I greatly appreciated this paper and believe it advances the field. I do believe there are a few shortcomings that could be elaborated upon to make the manuscript much stronger and provide more value for further research, but nothing so much as to question the benefit or validity of the work presented.
Title: Suggest altering to “Case-Control study: endogenous procalcitonin and protein carbonylated content as potential biomarkers of subclinical mastitis in dairy cows”. This reflects the mixed results of the study more clearly in the title rather than proposing both measures as biomarkers.
Line 10-12: Something is off in the opening sentence. Please edit accordingly.
Lines 17-18 and 28-30: It is stated that PCT seems to be a valuable marker for mastitis. This 1) is incongruent with the statement made in the conclusion discussion of the potential for use as a biomarker, 2) likely overstates the results given a Se and Sp in the 60s, and 3) states only mastitis when it should be made clear this is subclinical mastitis focused.
Line 34-35: Again, the opening sentence is very awkward. Unclear what is being stated.
Line 47: States all previous studies were in bovids when some were in equids.
Line 48-49: What deep knowledge exists in human medicine is not made clear, nor are references provided.
Line 65-66: Please be cautious here and elsewhere when discussing oxidative stress. Oxidative stress is not a given during disease. It occurs only when antioxidant defenses are overcome, not simply because reactive oxygen species are increased.
Line 70: I’m not confident that laboratory tests are necessary to detect subclinical mastitis. Many farms diagnose this on farm.
Line 119: Here it states all quarters were sampled. The opening sentence of this paragraph to me reads as though a single quarter was sampled. Please clarify.
Line 120: Given the emphasis in your discussion on the impact finding an etiologic agent of the mastitis can have on results, I would appreciate knowing what assay(s) were used to characterize the types of pathogens present.
Line 128/Section 2.5: I would appreciate seeing the methods for the PCC assay described more clearly rather than simply reference another work. Given the interesting results with this data, I believe it is important to state clearly how samples were assessed since it is not a kit.
175-177: Here or elsewhere please explain how, without a CBC/Chem or tests for other markers of inflammation, other subclinical diseases cannot be ruled out that could skew results.
Line 196: With a significant proportion of unhealthy cows at 3+ lactations, please describe how parity number could impact your results. The data is presented and there appears to be an imbalance but it is not discussed how this may impact your data.
Line 197: Please describe in the methods how TP was measured (unless I missed it) and include why as it is only briefly mentioned in the discussion.
Line 224-225: I rather disagree with the statement that the data for PCT show an acceptable level of diagnostic accuracy. Se and Sp in the 60s would not be appropriate to drive decision-making on farm. It warrants further study, but little can be based on the threshold determined in this study alone.
Line 231-234: Please go into this deeper because there are exponential differences between subclinical mastitis in reference 19 and your data. Other references are discussed in the following paragraph but not this one.
Line 257-258: Suggest saying “when there is an identifiable bacterial infection present.” Some bacterial infections are unlikely to be identified even when present.
Line 269-277: I am not overly comfortable with this paragraph. I don’t believe this data could drive antibiotic decision making and with the Se and Sp as it is, it could easily lead to mis-use of antibiotics as well, especially if one is blindly treating based on these parameters alone without regard for etiologic agent. Suggest removal of One Health discussion.
Line 291-303: If I can offer a suggestion or potential explanation, I would consider your differences in DIM as a confounding variable. Did you run stats to see if there is a significant difference between the groups regarding DIM? I would expect you to find one. Cows generally early in lactations have been shown to face greater oxidative stress in the first third of lactation than mid-lactation animals. I believe the Sordillo lab has a couple of papers on this and there are likely others in the literature as well. This simple difference in lactation stage could explain differences in PCC, ie. Healthy cows are earlier in lactation and could higher higher oxidative stress markers because of this while unhealthy cows are further along in lactation and thus have a lower level of systemic reactive oxygen species.
Line 311: I like this phrasing of how to interpret the data much better than the abstract.
Line 311: I don’t fully understand how such a diagnostic tool would practically contribute to antibiotic decision-making nor why someone would choose this assay rather than somatic cell count, which it seems would provide 100% accuracy. Maybe better discussed more thoroughly in the Discussion.
Comments on the Quality of English LanguageThere are a couple of sentences that need to be looked at such as the title, first sentence of the short abstract, and first sentence of the intro, but generally, it is fine.
Author Response
Thank you for the opportunity to review the manuscript by Sala et al. Case-Control study: endogenous procalcitonin and protein carbonylated content as a biomarker of subclinical mastitis in dairy cow. This manuscript compares the plasma values of PCT and PCC in both healthy cows and those with sub-clinical mastitis across parities and periods of lactation. The findings were positive for PCT and negative for PCC, compared to the proposed hypothesis, but it is VERY appreciated that the negative data was presented thoroughly and discussed.
Overall, I greatly appreciated this paper and believe it advances the field. I do believe there are a few shortcomings that could be elaborated upon to make the manuscript much stronger and provide more value for further research, but nothing so much as to question the benefit or validity of the work presented.
Au: We want to express our thanks to the reviewer, whose comments have contributed to improve our paper.
Title: Suggest altering to “Case-Control study: endogenous procalcitonin and protein carbonylated content as potential biomarkers of subclinical mastitis in dairy cows”. This reflects the mixed results of the study more clearly in the title rather than proposing both measures as biomarkers.
Au: thanks for the suggestion. The title was changed followed your suggestion.
Line 10-12: Something is off in the opening sentence. Please edit accordingly.
Au: sorry for the mistake. The sentence was corrected.
Lines 17-18 and 28-30: It is stated that PCT seems to be a valuable marker for mastitis. This 1) is incongruent with the statement made in the conclusion discussion of the potential for use as a biomarker, 2) likely overstates the results given a Se and Sp in the 60s, and 3) states only mastitis when it should be made clear this is subclinical mastitis focused.
Au: the conclusion in the abstract was changed following your suggestion. Lines 28-29.
Line 34-35: Again, the opening sentence is very awkward. Unclear what is being stated.
Au: again, sorry for the mistake. The sentence was corrected.
Line 47: States all previous studies were in bovids when some were in equids.
Au: The change was made. Line 71.
Line 48-49: What deep knowledge exists in human medicine is not made clear, nor are references provided.
Au: thanks for the comment. The paragraph about PCT was expanded, incorporating the additional information requested. See at lines 42-67.
Line 65-66: Please be cautious here and elsewhere when discussing oxidative stress. Oxidative stress is not a given during disease. It occurs only when antioxidant defenses are overcome, not simply because reactive oxygen species are increased.
Au: Thanks for the suggestion. The sentence was rewrite. Lines 73-81.
Line 70: I’m not confident that laboratory tests are necessary to detect subclinical mastitis. Many farms diagnose this on farm.
Au: Thank you for the comment. The previous sentence likely caused confusion. We have revised it to clarify that subclinical mastitis is not easily diagnosed due to the absence of macroscopic alterations in the milk, requiring additional diagnostic methods even under field conditions. See lines 92-107.
Line 119: Here it states all quarters were sampled. The opening sentence of this paragraph to me reads as though a single quarter was sampled. Please clarify.
Au: Thank you for the comment. Each milk quarter of the cows included in the study was analyzed to identify which quarter was affected by SCM and to ensure that all quarters of cows in the H group were healthy. The word "single" at line 233 has been removed.
Line 120: Given the emphasis in your discussion on the impact finding an etiologic agent of the mastitis can have on results, I would appreciate knowing what assay(s) were used to characterize the types of pathogens present.
Au: Thanks for the suggestion. The method used to bacteriological analysis was added at lines 241-248.
Line 128/Section 2.5: I would appreciate seeing the methods for the PCC assay described more clearly rather than simply reference another work. Given the interesting results with this data, I believe it is important to state clearly how samples were assessed since it is not a kit.
Au: Thanks for the suggestion. The method used for the PCC analysis was rewrite with more detail at lines 264-301.
175-177: Here or elsewhere please explain how, without a CBC/Chem or tests for other markers of inflammation, other subclinical diseases cannot be ruled out that could skew results.
Au: Thanks for the comment. We added this information in the limitation of the study. See lines 742-747.
Line 196: With a significant proportion of unhealthy cows at 3+ lactations, please describe how parity number could impact your results. The data is presented and there appears to be an imbalance, but it is not discussed how this may impact your data.
Line 291-303: If I can offer a suggestion or potential explanation, I would consider your differences in DIM as a confounding variable. Did you run stats to see if there is a significant difference between the groups regarding DIM? I would expect you to find one. Cows generally early in lactations have been shown to face greater oxidative stress in the first third of lactation than mid-lactation animals. I believe the Sordillo lab has a couple of papers on this and there are likely others in the literature as well. This simple difference in lactation stage could explain differences in PCC, ie. Healthy cows are earlier in lactation and could higher higher oxidative stress markers because of this while unhealthy cows are further along in lactation and thus have a lower level of systemic reactive oxygen species.
Au: Thank you for your comments. We have added statistical analyses (see Materials and Methods and Results) to investigate whether different cow parity and DIM could influence PCC and PCT. However, despite a difference in the two groups for these variables, they do not impact the two biomarkers. This result has also been included in the discussions. See lines 732-738.
Line 197: Please describe in the methods how TP was measured (unless I missed it) and include why as it is only briefly mentioned in the discussion.
Au: Thank you for the comment. The method for determining TP has been explained in more detail. The TP measured using the Lowry method is functional for calculating PCC but not comparable to TP used in clinical practice. Therefore, the authors have chosen to retain the description of the TP determination method as it is essential for calculating PCC, but they have excluded it from the results as it may cause confusion and cannot be discussed with TP used as a biomarker.
Line 224-225: I rather disagree with the statement that the data for PCT show an acceptable level of diagnostic accuracy. Se and Sp in the 60s would not be appropriate to drive decision-making on farm. It warrants further study, but little can be based on the threshold determined in this study alone.
Au: Thanks for the suggestion. The “acceptable level of diagnostic accuracy” was removed.
Line 231-234: Please go into this deeper because there are exponential differences between subclinical mastitis in reference 19 and your data. Other references are discussed in the following paragraph but not this one.
AU: thanks for the comment. The discussion of this finding was added at lines 488-490.
Line 257-258: Suggest saying “when there is an identifiable bacterial infection present.” Some bacterial infections are unlikely to be identified even when present.
Au: Thanks. The sentence was change followed you suggestion.
Line 269-277: I am not overly comfortable with this paragraph. I don’t believe this data could drive antibiotic decision making and with the Se and Sp as it is, it could easily lead to mis-use of antibiotics as well, especially if one is blindly treating based on these parameters alone without regard for etiologic agent. Suggest removal of One Health discussion.
Au: This part of discussion was removed.
Line 311: I like this phrasing of how to interpret the data much better than the abstract.
Au: the same conclusion was reported in the abstract.
Line 311: I don’t fully understand how such a diagnostic tool would practically contribute to antibiotic decision-making nor why someone would choose this assay rather than somatic cell count, which it seems would provide 100% accuracy. Maybe better discussed more thoroughly in the Discussion.
Au: The conclusion of the study was revised based on your indication. The potential use of PCT as a tool for starting of antibiotic therapy is extensively studied in human medicine, and the authors foresee a similar application of PCT in bovine medicine. Specifically, in human medicine, PCT values differ between infections caused by Gram-positive and Gram-negative bacteria. This would be highly interesting in the field of bovine medicine but given the current state of knowledge in veterinary medicine regarding PCT, it might be premature to delve into the discussion and, most importantly, draw conclusions on this topic. Thank you for your comment.
Comments on the Quality of English Language:
There are a couple of sentences that need to be looked at such as the title, first sentence of the short abstract, and first sentence of the intro, but generally, it is fine.
Au: the corrections were made.
Reviewer 3 Report
Comments and Suggestions for Authors
In this paper, Sala and colleagues analyzed investigated procalcitonin and protein carbonylated content during subclinical mastitis in dairy cows and discovered a significant difference in PCT levels between healthy and subclinical mastitis cows.
The field is worthy of investigation. However, some text revision should be made:
In line 11 deleted dot (in dairy cows.are).
In line 24 for PCC cite methods, not "as described in literature".
In introduction requires additional backgournd regarding the diagnostic methods of clinical and subclinical mastitis.
My major treatments concernes are:
Why did the authors decide to measure PCT and PCC from plasma and not from milk?
Why did you include only cows with subclinical mastitis and not cows with clinical mastitis?
Author Response
In this paper, Sala and colleagues analyzed investigated procalcitonin and protein carbonylated content during subclinical mastitis in dairy cows and discovered a significant difference in PCT levels between healthy and subclinical mastitis cows.
Au: We want to express our thanks to the reviewer, whose comments have contributed to enhancing our paper.
The field is worthy of investigation. However, some text revision should be made:
In line 11 deleted dot (in dairy cows.are).
Au: sorry for the mistake. The change was made.
In line 24 for PCC cite methods, not "as described in literature".
Au: Thanks for suggestion. The change was made at line 24.
In introduction requires additional backgournd regarding the diagnostic methods of clinical and subclinical mastitis.
Au: Thanks for the suggestion. The paragraph regarding mastitis has been modified following your suggestions and those of other reviewers. See lines 97-107.
My major treatments concernes are:
Why did the authors decide to measure PCT and PCC from plasma and not from milk?
Why did you include only cows with subclinical mastitis and not cows with clinical mastitis?
Au: Thanks for the question. The decision to measure PCT in plasma rather than milk is based on a prior study conducted by our research group, where the kit we utilized was found to be incapable of identifying PCT in milk (Meucci, V., Orsetti, C., Sgorbini, M., Battaglia, F., Cresci, M., & Bonelli, F. (2022). Can Procalcitonin Be Dosed in Bovine Milk Using a Commercial ELISA Kit?. Animals, 12(3), 289). Consequently, the choice was made to measure PCC exclusively in plasma as well. Regarding the decision to focus only on subclinical mastitis, this choice was made because, from a clinical perspective of biomarker utilization, we believe it is more pertinent to apply them in subclinical mastitis. In cases of clinical mastitis, diagnosis is straightforward, and ancillary methods are crucial for determining the involved pathogens, where biomarkers currently play a marginal role.
Round 2
Reviewer 1 Report
Comments and Suggestions for Authors
The manuscript has been sufficiently improved according to the comments and I propose it for publication.